

# Grassland productivity in response to nutrient additions and herbivory is scale-dependent

Erica A.H. Smithwick[1], Douglas C. Baldwin[2] and Kusum J. Naithani[3]

[1] Department of Geography and Intercollege Graduate Degree Program in Ecology, Pennsylvania State University, University Park, PA, United States
[2] Department of Geography, Pennsylvania State University, University Park, PA, United States
[3] Department of Biological Sciences, University of Arkansas, Fayetteville, AR, United States

## ABSTRACT

Vegetation response to nutrient addition can vary across space, yet studies that explicitly incorporate spatial pattern into experimental approaches are rare. To explore whether there are unique spatial scales (grains) at which grass response to nutrients and herbivory is best expressed, we imposed a large (∼3.75 ha) experiment in a South African coastal grassland ecosystem. In two of six 60 × 60 m grassland plots, we imposed a scaled sampling design in which fertilizer was added in replicated sub-plots (1 × 1 m, 2 × 2 m, and 4 × 4 m). The remaining plots either received no additions or were fertilized evenly across the entire area. Three of the six plots were fenced to exclude herbivory. We calculated empirical semivariograms for all plots one year following nutrient additions to determine whether the scale of grass response (biomass and nutrient concentrations) corresponded to the scale of the sub-plot additions and compared these results to reference plots (unfertilized or unscaled) and to plots with and without herbivory. We compared empirical semivariogram parameters to parameters from semivariograms derived from a set of simulated landscapes (neutral models). Empirical semivariograms showed spatial structure in plots that received multi-scaled nutrient additions, particularly at the 2 × 2 m grain. The level of biomass response was predicted by foliar P concentration and, to a lesser extent, N, with the treatment effect of herbivory having a minimal influence. Neutral models confirmed the length scale of the biomass response and indicated few differences due to herbivory. Overall, we conclude that interpretation of nutrient limitation in grasslands is dependent on the grain used to measure grass response and that herbivory had a secondary effect.

## INTRODUCTION

Nutrient limitation is known to constrain ecosystem productivity (*Vitousek & Howarth, 1991*; *LeBauer & Treseder, 2008*; *Fay et al., 2015*). In general, temperate systems are expected to have greater levels of nitrogen (N) limitation on vegetation growth than sub-tropical or tropical systems, where phosphorus (P) may be more limiting due to highly weathered soils (*Vitousek & Sanford, 1986*; *Hedin, 2004*; *Lambers et al., 2008*; *Domingues et al., 2010*),

Corresponding author
Erica A.H. Smithwick,
smithwick@psu.edu

although co-limitation and the role of other nutrients is also acknowledged to be important (*Fay et al., 2015*). Ecological inference is dependent on the observational scale of the measurements, however, and as such, our ability to infer ecosystem function from patterns in nutrient availability rests on the grain and extent of the measurement (*Dungan et al., 2002*). In a spatial context, grain reflects the finest level of resolution (precision of measurement) whereas extent refers to the size of the study area, and the choice of these dimensions offer differ among studies (*Turner & Gardner, 2015*). In the case of nutrient limitation, the optimal grain for diagnosing nutrient limitation, especially in grassland ecosystems, is not known and may vary at fine scales (*Klaus et al., 2016*). Patchiness in nutrient availability can be governed by variability in soil properties or terrain, spatial variability in microbial community composition, or differential nutrient affinities across functional groups that have different spatial or temporal distributions (*Reich et al., 2003*; *Ratnam et al., 2008*). Perhaps as a result of this spatial heterogeneity, N, P, and N + P limitations on vegetation productivity have all been documented in African savanna or grassland systems (*Augustine, McNaughton & Frank, 2003*; *Craine, Morrow & Stock, 2008*; *Okin et al., 2008*; *Ngatia et al., 2015*). This study asks whether new approaches that actively test (sensu *McIntire & Fajardo, 2009*) the scale of grass response to nutrients and herbivory can aid understanding of nutrient limitation in grassland ecosystems.

Herbivores influence nutrient availability and can further enhance or diminish spatial and temporal variability in nutrient limitation (*Senft et al., 1987*; *Robertson, Crum & Ellis, 1993*; *Augustine & Frank, 2001*; *Okin et al., 2008*; *Liu et al., 2016*). Herbivores affect spatial patterns of nutrient availability directly through deposition of nutrient-rich manure or urine, which can lead to heterogeneous patterns of primary productivity (*Fuhlendorf & Smeins, 1999*). As animals move across an area and rest in new locations, variability can be further enhanced (*Auerswald, Mayer & Schnyder, 2010*; *Fu et al., 2013*). On the other hand, consumption of nutrient-rich grasses may reduce overall variance by reducing differences in biomass amounts compared to ungrazed areas. Through model simulations, *Gil, Jiao & Osenberg (2016)* recently showed that herbivores may have a greater influence on controlling biomass at fine versus broad extents, suggesting scale-dependence in herbivore control of plant biomass. In a field experiment, *Van der Waal et al. (2016)* concluded that herbivore consumption of nutrient rich patches eliminated the positive effects of fertilization on the plant community and that patchiness itself (independent of the patch size) can affect the outcome of trophic relationships in grassland and savanna ecosystems. Taken together, understanding scale dependence (*Sandel, 2015*), specifically the degree to which grass productivity is governed by the grain and extent nutrient availability and herbivore activity, is important for making inferences about ecosystem function in grasslands and requires new methodological approaches for its study.

Incorporating spatial autocorrelation into ecological studies has augmented our understanding of how spatial structure of soils, plants, and climate regulates ecosystem function, often at multiple, nested scales (*Watt, 1947*; *Turner, Donato & Romme, 2013*). Understanding the autocorrelation structure of key ecosystem properties is critical for determining optimal scales for studying ecological systems, interpreting change in ecological communities, and assessing landscape connectivity or ecosystem resilience. However, for

any given study, the scale of this autocorrelation structure and its implications for inferring ecological processes are not known in advance. Select studies have employed experimental spatial designs *a priori* (*Stohlgren, Falkner & Schell, 1995*) or have used computational simulations to explore the influence of space on ecosystem properties (*With & Crist, 1995*; *Smithwick, Harmon & Domingo, 2003*; *Jenerette & Wu, 2004*). Geostatistical analysis is commonly used (*Jackson & Caldwell, 1993b*; *Robertson, Crum & Ellis, 1993*; *Smithwick et al., 2005*; *Jean et al., 2015*) to describe the grain and extent of observed ecological patterns, while other approaches may be more useful for predictive modeling of ecological processes through space and time (*Miller, Franklin & Aspinall, 2007*; *Beale et al., 2010*), though these, too, rest on an understanding of autocorrelation structures.

Understanding these spatial structures is often elusive because ecological patterns develop from complex interactions among individuals across variable abiotic gradients (*Jackson & Caldwell, 1993a*; *Rietkerk et al., 2000*; *Ettema & Wardle, 2002*) and manifest at multiple spatial scales (*Mills et al., 2006*). Disturbances further create structural patterns that may influence ecological processes at many scales (*Turner et al., 2007*; *Schoennagel, Smithwick & Turner, 2008*). Resultant patchiness in ecological phenomena is common. For example, *Rietkerk et al. (2000)* observed patchiness in soil moisture at three unique scales (0.5 m, 1.8 m and 2.8 m) in response to herbivore impacts. Following fire in the Greater Yellowstone Ecosystem (Wyoming, USA), *Turner et al. (2011)* observed variation in soil properties at the level of individual soil cores, and *Smithwick et al. (2012)* observed autocorrelation in post-fire soil microbial variables that ranged from 1.5 to 10.5 m. Patchiness in soil resources at the level of individual shrubs and trees has been demonstrated by several studies (*Liski, 1995*; *Pennanen et al., 1999*; *Hibbard et al., 2001*; *Lechmere-Oertel, Cowling & Kerley, 2005*; *Dijkstra et al., 2006*). In savanna systems, multiple spatial scales are needed to explain complex grass-tree interactions (*Mills et al., 2006*; *Okin et al., 2008*; *Wang et al., 2010*; *Pellegrini, 2016*) and it is likely that these factors are nested hierarchically with spatial scale (*Pickett, Cadenasso & Benning, 2003*; *Rogers, 2003*; *Pellegrini, 2016*).

In the absence of understanding the scale at which ecosystems are nutrient-limited, nor the causal mechanisms underlying this scale-dependence, the ability to extrapolate nutrient limitations to broader areas is hindered. Here we report on a study in which we tested the grain-dependence of grass biomass to nutrient additions and herbivory using a novel experimental design. Our objectives were to: (1) quantify the grain size at which vegetation biomass and nutrient concentrations respond to nutrient additions in fenced and unfenced plots, (2) relate the level of biomass response to plant nutrient concentrations and herbivory and (3) assess the degree to which herbivory and nutrient treatments explained the spatial structure of grass productivity through comparison of empirical semivariograms and neutral models (simulated semivariogram models based on prescribed landscape patterns).

For Objective 1, we hypothesized that the grass response would differ between three subplots scales at which fertilizer was added (1 × 1 m, 2 × 2 m, and 4 × 4 m). These patch sizes were chosen to correspond to ecosystem processes that might govern nutrient uptake, including the spacing of individual plants, plant groupings, or plot-level topography, respectively, which have been identified as critical sources of variation in

soil biogeochemistry (*Jackson & Caldwell, 1993a*; *Rietkerk et al., 2000*; *Ettema & Wardle, 2002*). We posited that, at the finest sampling grain (1 × 1 m), grass biomass and nutrient concentrations would likely reflect competition for nutrient resources among individuals of a given species, or between occupied and unoccupied (open) locations (*Remsburg & Turner, 2006*; *Horn et al., 2015*). At the intermediate grain (2 × 2 m), we expected that biomass and nutrient concentrations would reflect the outcome of competitive exclusion among grass clumps comprised of different species (*Grime, 1973*; *Schoolmaster, Mittelbach & Gross, 2014*; *Veldhuis et al., 2016*). At the largest grain (4 × 4 m), we expected that abiotic processes such as variability in hydrology or soil properties would strongly determine the response of grass biomass and nutrient concentrations in addition to competitive processes among individuals and species (*Ben Wu & Archer, 2005*; *Mills et al., 2006*). Half of the plots were fenced to exclude herbivory to determine whether there were differences the scale of the response due to animal activity. We used a semivariogram model developed from empirical data and used model parameters to estimate the spatial structure of biomass and nutrient concentrations. We expected that biomass and vegetation nutrient concentrations would have range parameters from empirical semivariograms that corresponded to the hypotenuse distances of the subplot scales (i.e., 1 m, 2.83 m, and 5.66 m hypotenuse distances for the 1 × 1 m, 2 × 2 m, and 4 × 4 m subplots, respectively). We expected that patchiness would be highest. i.e., range scales would be smaller, for the unfenced, heterogeneously fertilized plot because these areas would have received nutrient additions in the form of manure and urine from animal activity in addition to nutrient additions (*Liu et al., 2016*)

For Objective 2, we hypothesized that biomass responses to nutrient additions at the plot level would best explained by foliar N and P concentrations, given previous work indicating the importance of coupled nutrient limitation to grassland productivity (*Craine, Morrow & Stock, 2008*; *Craine & Jackson, 2010*; *Ostertag, 2010*; *Fay et al., 2015*). We expected that herbivory would have limited effects on biomass productivity relative to the influence of nutrients.

To test the robustness of our empirical results against a broader set of prescribed landscape patterns (Objective 3), we compared the empirical semivariogram models with neutral semivariogram models, based on computer-simulated landscapes that mimic hypothesized patterns due to known ecological processes (*Fajardo & McIntire, 2007*). This approach allowed us to compare empirical patterns across a set of null models in which the patterns were known. We also could avoid issues of pseudoreplication associated with the limited set of replications in the field by developing a set of artificial landscapes in which we imposed known herbivory and nutrient patterns. Using this approach, the null assumption is that ranges (autocorrelation distances, or length scales) calculated in the neutral models would be similar to the ranges calculated from empirical data. Similarity of model parameters between empirical and neutral models would provide confidence that observed patterns reflect known ecological processes. We hypothesized that that there would be greater spatial structure in plots that received heterogeneous fertilizers compared to reference plots. In homogenously fertilized plots or unfertilized plots, spatial structure would be observed at scales other than scales of the subplots (or not at all) and we would

expect to see lower levels of spatial structure explained by the model relative to random processes (higher nugget:sill, described below).

## METHODS

### Study area

This study was conducted in Mkambathi Nature Reserve, a 7720-ha protected area located at 31°13′27″S and 29°57′58″E along the Wild Coast region of the Eastern Cape Province, South Africa. The Eastern Cape is at the confluence of four major vegetative groupings (Afromontane, Cape, Tongaland-Pondoland, and Karoo-Namib) reflecting biogeographically complex evolutionary histories. It is located within the Maputaland-Pondoland-Albany conservation area, which bridges the coastal forests of Eastern Africa to the north, and the Cape Floristic Region and Succulent Karoo to the south and west. The Maputaland-Pondoland-Albany region is the second richest floristic region in Africa, with over 8,100 species identified (23% endemic), and 1,524 vascular plant genera (39 endemic) (*CEPF, 2010*) Vegetation in Mkambathi is dominated by coastal sour grassveld ecosystems, which dominate about 80% of the ecosystem (*Shackleton et al., 1991*; *Kerley, Knight & DeKock, 1995*), with small pockets of forest along river gorges, wetland depressions, and coastal dunes. Dominant grasses in the Mkambathi reserve include the coastal *Themeda triandra—Centella asiatica* grass community, the tall grass *Cymbopogon validas—Digitaria natalensis* community in drier locations, and the short-grass *Tristachya leucothrix-Loudetia simplex* community (*Shackleton, 1990*). Grasslands in Mkambathi have high fire frequencies, and typically burn biennially. Soils are generally derived from weathered Natal Group sandstone and are highly acidic and sandy with weak structure and soil moisture holding capacity (*Shackleton et al., 1991*).

Annual precipitation in Mkambathi Reserve averaged 1,165 mm yr$^{-1}$ between 1925 and 2015 and 1,159 mm yr$^{-1}$ between 2006 and 2015. June is typically the driest month (averaging 30.8 mm 1996–2015) and March is typically the wettest month (averaging 147.6 mm 1996–2015). For nearby Port Edward, South Africa, where data was available, the maximum temperature is highest in February (26.7 °C), averaging 23.7 °C annually, while minimum temperature is coolest in July (average 13.0 °C), averaging 17.4 °C annually. During the years of this study (2010–2012), annual temperature averaged 17.4 °C (min) to 23.7 °C (max), well within the historical average. The year 2010 was one of the driest years on record (656.6 mm yr$^{-1}$), whereas 2011 and 2012 (1413.6 and 1766.3 mm yr$^{-1}$ respectively) were wetter years than average, although within the historical range (652.8–2385.9 mm yr$^{-1}$). All climate data were obtained from the South African Weather Service.

### Nutrient addition experiment

We established a large-scale experimental site that included six 60 × 60 m plots arranged in a rectangular grid (Eastern Cape Parks and Tourism Agency Permit RA0081). The site was surrounded by a fuel-removal fire-break and each plot was separated by at least 10 m for a total size of 3.75 ha for the entire site. To account for grazing, a fence was constructed around three of these plots to exclude herbivores. Nutrient additions were applied to four
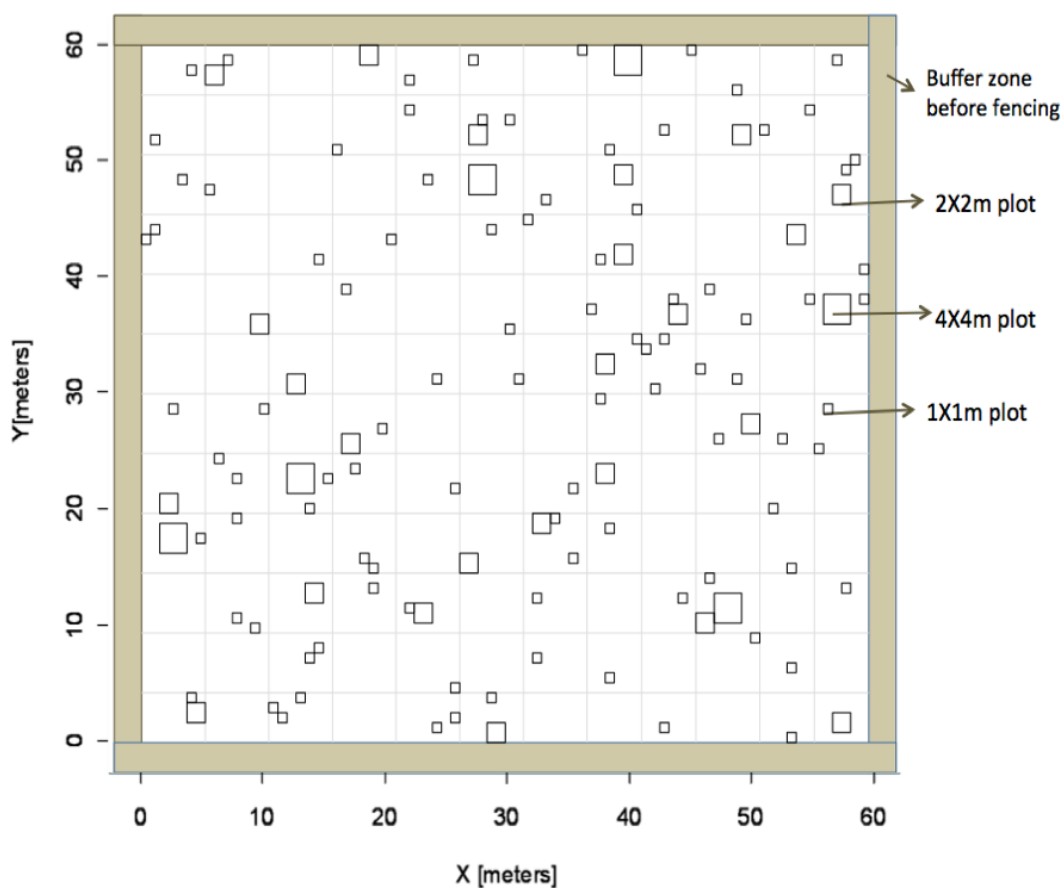

**Figure 1 Experimental design.** Overview of experimental design based on Latin Hypercube sampling used to identify subplot locations to receive fertilizer in the heterogeneous plots.

plots whereas two plots received no fertilizer additions; plot treatment was random. Of the four plots that received fertilization, two received nutrients evenly across the entire $60 \times 60$ m plot ("homogenous plots") and the other two fertilized plots received nutrient additions within smaller subplots in a heterogeneous design ("heterogeneous plots"). Within heterogeneous plots, fertilizer was applied within subplots of three different sizes ($1 \times 1$ m, $2 \times 2$ m, and $4 \times 4$ m) that were replicated randomly across each plot (Fig. 1). Location of individual subplots was determined prior to field work using a Latin Hypercube random generator that optimizes the variability of lag distances among sampling plots and is ideal for geostatistical analysis (*Xu et al., 2005*). There were a total of 126 subplots that received fertilizer in each heterogeneous plot. All sampling locations were geo-referenced with a GPS (Trimble 2008 Series GeoXM; 1 m precision) and flagged. The number of sub-plot units at each scale was determined so as to equalize the total fertilized area at each sub-plot scale (i.e., six $4 \times 4$ m plots and 24, $2 \times 2$ m plots). To ensure aboveground grass biomass would respond to nutrient additions, we employed a dual (nitrogen (N) + phosphorus (P)) nutrient addition experiment. Additional N was added as either ammonium nitrate (230 g kg$^{-1}$ N) or urea (460 g kg$^{-1}$) at a rate of 10 g m$^{-2}$ yr$^{-1}$ in a single application, following the protocols of *Craine, Morrow & Stock (2008)*.

Additional P was added as superphosphate (105 g kg$^{-1}$ P) at a rate of 5 g m$^{-2}$ yr$^{-1}$. Dual addition (N + P) was chosen to increase the likelihood of treatment response and increase geostatistical power by reducing the number of treatments, thus increasing sample size. Towards the end of the summer wet season (February), we applied fertilizer to subplots in the two heterogeneous plots and evenly across the two homogeneous plots. The amount of fertilizer received was equal on a per unit area basis among plots and subplots.

## Vegetation and soil sampling

One year following nutrient additions, a subset of subplots was sampled for soil and vegetation nutrient concentrations and biomass. Subplots to be sampled were selected randomly prior to being in the field using the Latin Hypercube approach. The approach allowed us to specify a balanced selection of subplots within each subplot size class (four 4 × 4 m, eight 2 × 2 m, and thirty-two 1 × 1 m). Within each subplot that was revisited, we randomly selected locations for biomass measurement and vegetation clippings: two locations were identified and flagged from within the 1 × 1 m subplots (center coordinate and a random location 0.5 m from center), four samples were identified and flagged from within the 2 × 2 m subplots, and eight samples were identified and flagged from within the 4 × 4 m subplots.

At each flagged location within sampled subplots, productivity was measured as grass biomass using a disc pasture meter (DPM) (*Bransby & Tainton, 1977*) and grab samples of grass clippings were collected for foliar nutrient analysis, using shears and cutting to ground-level. Calibration of the DPM readings was determined using ten random 1 × 1 m subplots in each plot ($n = 60$ total) that were not used for vegetation or soil harvesting, in which the entire biomass was harvested to bare soil. Linear regression was used to relate DPM estimates with harvested biomass at calibration subplots ($R^2 = 0.76$, $p < 0.0001$; Fig. S1) and the resulting equation was then used to estimate biomass at the remaining 606 locations.

Soil samples from the top 0–10 cm soil profile depth were collected adjacent to vegetation samples. Due to logistical and financial constraints, these samples were collected in fenced plots only. The A horizon of the Mollisols was consistently thicker than 10 cm, so all samples collected were drawn from the A horizon. Soil samples were shipped to BEMLab (Strand, South Africa) for nutrient analysis.

## Laboratory analysis

Biomass samples were separated into grasses and forbs, weighed, dried for 24 h at 60 °C, and reweighed. Vegetation nutrient samples were dried, ground with a 40 mm grinding mesh, and then shipped to the Penn State Agricultural Analytical Laboratory (University Park, Pennsylvania; USDA Permit PDEP11-00029). Grass P concentration was analyzed using a hot block acid digestion approach (*Huang & Schulte, 1985*) and grass N concentration was measured with a Combustion-Elementar Vario Max method (*Horneck & Miller, 1998*). Soil N and C concentrations were determined on a LECO elemental analyzer (Leco Corporation, St. Joseph, MI). Soil P was analyzed using acid extraction following the method of *Wolf & Beegle (1995)*. Soil pH was estimated using KCl extraction following *Eckert & Sims (1995)*.

**Table 1  Plot-level biomass and vegetation nutrient concentrations.** Mean (±1 standard error (SE)) biomass, vegetation N concentration, vegetation P concentration, and N:P ratios across experimental plots in Mkambathi Nature Reserve, one year following nutrient fertilization.

| Treatment | Average Biomass ± 1 SE (g m$^{-2}$) | Average N ± 1 SE (%) | Average P ± 1 SE (%) | N:P | n |
|---|---|---|---|---|---|
| Fenced | | | | | |
|     Unfertilized | 411.9 ± 9.75 | 0.646 ± 0.024 | 0.036 ± 0.001 | 17.9 | 134 |
|     Heterogeneous | 542.4 ± 15.05 | 0.747 ± 0.041 | 0.048 ± 0.002 | 15.6 | 120 |
|     Homogeneous | 456.2 ± 8.28 | 0.710 ± 0.014 | 0.054 ± 0.002 | 13.2 | 117 |
| Unfenced | | | | | |
|     Unfertilized | 483.6 ± 13.70 | 0.576 ± 0.011 | 0.038 ± 0.001 | 15.2 | 132 |
|     Heterogeneous | 562.6 ± 18.60 | 0.775 ± 0.015 | 0.064 ± 0.002 | 12.1 | 128 |
|     Homogeneous | 375.4 ± 5.96 | 0.722 ± 0.017 | 0.059 ± 0.002 | 12.2 | 124 |

## Empirical semivariograms

Semivariogram models were fit to empirical data and model parameters were used to test Objective 1. The range parameter was used to estimate the scale of autocorrelation; the sill parameter was used to estimate overall variance; and the nugget parameter was used to represent variance not accounted for in the sampling design. A maximum likelihood approach was used to quantify the model parameters. This approach assumes that the data $(Y_1 \ldots Y_n)$ are realizations of an underlying spatial process, and that the distribution of the data follows a Gaussian multivariate distribution:

$$Y \sim N(\mu 1, C\Sigma + C_0 I) \tag{1}$$

where $\mu$ is the mean of the data multiplied by an n-dimensional vector of 1's, $C$ is the partial sill (total sill $= C_0 + C$), $\Sigma$ is an n × n spatial covariance matrix, $C_0$ is the nugget effect, and $I$ is an n × n identity matrix. The $i$, $j$th element of $\Sigma$ is calculated with a spatial covariance function $\rho(h_{ij})$, where $h_{ij}$ is the Euclidean distance between measurement points $i$ and $j$. An exponential covariance model was chosen for its relative simplicity. The full equation for summarizing the second order moment for an element $i$, $j$ is:

$$\gamma(h_{ij}) = C_0 + C\left[\exp\left(\frac{-h_{ij}}{\phi}\right)\right] \tag{2}$$

where $\gamma(h_{ij})$ is the modeled spatial covariance for measurements $i$ and $j$, $\phi$ is the range parameter, and $3 * \phi$ is the range of spatial autocorrelation. The underlying spatial mean $\mu$ may be held constant or estimated with a linear model across all locations and in this case we used the plot-level mean of the data for $\mu$ (Table 1).

The measured soil and plant variables exhibited varying degrees of non-normality in their distributions, which violated the assumption of Gaussian stationarity within the underlying spatial data generating process. To uphold this assumption, we transformed variables at each plot using a box-cox transformation (*Box & Cox, 1964*):

$$\begin{aligned} \hat{Y}_i &= (Y_i^\lambda - 1)/\lambda \quad && if \ \lambda \neq 0 \\ \hat{Y}_i &= \log(Y_i) \quad && if \ \lambda = 0 \end{aligned} \tag{3}$$

where $Y_i$ is an untransformed variable (e.g., biomass) at location $i$, $\hat{Y}_i$ is the transformed variable, and $\lambda$ is a transformation parameter. We optimized the three spatial covariance model parameters and the transformation parameter $(C_0, C, \phi, \lambda)$ with the maximum likelihood procedure. A numerical finite-difference approximation algorithm selected the set of parameters that maximized a normal multivariate log-likelihood function (*Diggle, Ribeiro Jr & Christensen, 2003*). To approximate a sampling distribution of each parameter, a bootstrapping algorithm was used where a randomly sampled subset of data was input into the same maximum likelihood approach for 1,000 iterations. This provided a population of fitted parameters and models that was used to analyze the approximate distributions of each parameter for each plot. The maximum likelihood optimization was cross-validated by removing a random sub-sample of measurements from the optimization and then using the optimized model to make predictions at locations where measurements were removed. Observed vs. predicted values from the cross-validation procedure were then analyzed at each plot separately.

We used ordinary kriging (*Cressie, 1988*) with the optimized spatial covariance model from the maximum likelihood analysis to estimate biomass across all plots. Ordinary kriging is useful in this case because we detected spatial structure in the biomass data when considering all biomass data at once (see 'Results'). The geoR package (*Ribeiro Jr & Diggle, 2001*) in the R statistical language (*R Development Core Team, 2014*) was used for all spatial modeling and kriging.

### Mixed model

To relate these patterns in biomass to vegetation nutrient concentrations (Objective 2), we used a linear mixed modeling approach. Experimental factors such as herbivory, fertilizer type (i.e., heterogeneous, homogenous, and unfertilized), plot treatment, and subplot size were included as random effects to manage non-independence of data and avoid issues of pseudoreplication (*Millar & Anderson, 2004*). Multiple combinations of random effects and fixed effects were tested, where foliar N and P represented fixed effects upon biomass, and model error was assumed to be Gaussian. A normal likelihood function was minimized to estimate optimal regression coefficients for each mixed model formulation. To identify a mixed model that estimated biomass closely to observations, while also having the fewest possible parameters, we used the Akaike's Information Criterion (AIC) and Bayesian Information Criterion (BIC), which decrease with a negative log-likelihood function but increase with the number of parameters used in the model (*Burnham & Anderson, 2002*). The model with the lowest BIC was chosen as best representing the tradeoff of parsimony and prediction skill. The BIC associated with all other models was subtracted into the lowest available BIC, and models with a difference in BIC >2 were deemed significantly less favorable at estimating biomass and representing random effects than the model with the lowest BIC. All mixed modeling was conducted with the R package lme4.

### Simulated semivariograms

The neutral semivariogram models were constructed for six simulated landscapes (Fig. 2) to represent alternative landscape structures in response to nutrient addition and grazing: (Fig. 2A) fenced-unfertilized (biomass was assumed to be randomly distributed around

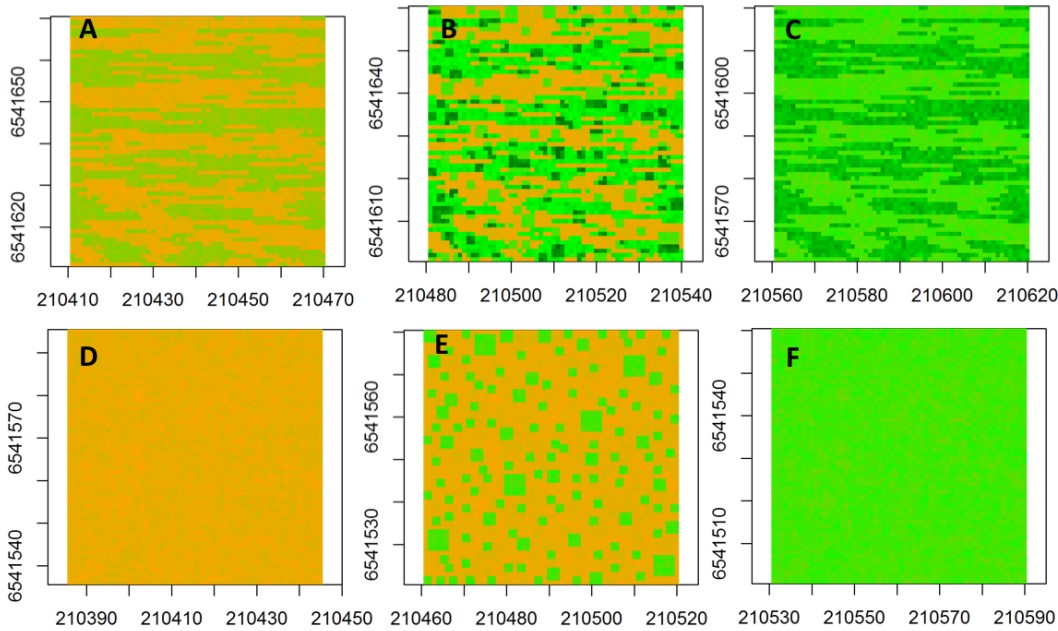

**Figure 2 Spatial maps of neutral models.** Spatial maps of neutral models used to simulate vegetation biomass for the following conditions: (A) Unfenced, unfertilized, (B) Unfenced, heterogeneously fertilized, (C) Unfenced-homogeneously fertilized, (D) Fenced, unfertilized, (E) Fenced, heterogeneously fertilized, (F) Fenced, homogeneously fertilized.

the mean of the biomass from the fenced, unfertilized experimental plot), (Fig. 2B) fenced-heterogeneous (biomass of Fig. 2A was doubled for selected subplots, following the same subplot structure that was used in the field experiments), (Fig. 2C) fenced-homogenous (biomass of Fig. 2A was doubled at every grid cell to mimic an evenly distributed fertilization response), (Fig. 2D) unfenced-unfertilized (biomass of Fig. 2A was increased by 50% in response to a combined effect of biomass loss by grazing and biomass gain by manure nutrient additions by herbivores; the increase occurred at a subset of sites to mimic random movement patterns of herbivores), (Fig. 2E) unfenced-heterogeneous (biomass equaled biomass of herbivory only, fertilizer only, or herbivory + fertilizer), and (Fig. 2F) unfenced-homogenous (biomass of Fig. 2D was doubled at all grid cells to mimic the additive effects of herbivores and homogenous fertilizer additions).

The spatial structure of simulated landscapes was analyzed using the same maximum likelihood approach as described for empirical models and data was not transformed. The mean ($\mu$) was estimated using a constant trend estimate. Given that the magnitude of observed and simulated biomass can change the amount of spatial variance, we scaled the nugget and sill parameters by dividing these parameters by the maximum calculated spatial autocorrelation in the data according to the 'modulus' method (*Cressie, 1993*).

## RESULTS

Vegetation biomass varied by 50% across plots, with the highest biomass found for heterogeneously fertilized plots (Table 1). Vegetation nutrient concentrations increased, and N:P ratios declined, following fertilization (Table 1). Vegetation N concentration

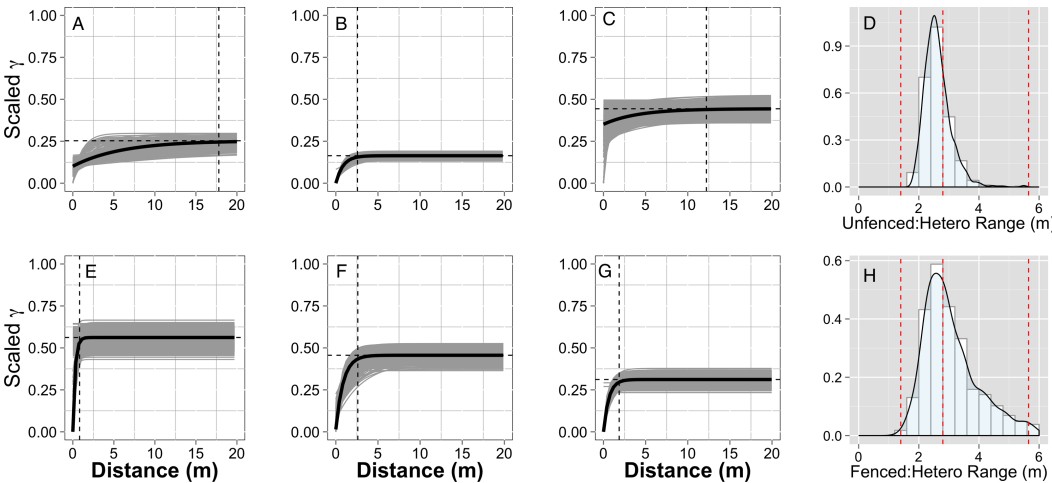

**Figure 3 Empirical semivariograms.** Empirical semi-variograms of vegetation biomass for each plot: (A) Unfenced, unfertilized, (B) Unfenced, Heterogeneously Fertilized, (C) Unfenced, homogeneously fertilized, (E) Fenced, unfertilized, (F) Fenced, heterogeneously fertilized, (G) Fenced, homogeneously fertilized. Shaded lines represent semi-variogram models fitted during the bootstrapping procedure. Dashed vertical line represents the range value. Also shown: the sampling distribution of the range parameter for heterogeneously fertilized plots that were either (D) Unfenced, or (H) Fenced. The distribution was calculated with a bootstrapping approach with maximum likelihood optimization. Dashed vertical lines represent the hypotenuses of the 1 × 1 m (1.4), 2 × 2 (2.8), and 4 × 4 (5.7) sub-plots.

averaged 0.60 ± 0.01% in unfertilized plots, 0.72 ± 0.02% in heterogeneously fertilized plots, and 0.77 ± 0.02% in homogenously fertilized plots, an increase of 20 % and 28%, respectively. Vegetation P concentration averaged 0.037 ± 0.001 mg g$^{-1}$ in unfertilized plots, 0.056 ± 0.002 mg g$^{-1}$ in heterogeneously fertilized, and 0.057 ± 0.002 mg g$^{-1}$ in homogeneously fertilized plots, an increase of 34 and 35%, respectively. The vegetation N:P ratios ranged from a high of 17.9 in the fenced-unfertilized plot to 12.1 in the unfenced-homogenously fertilized plot. Vegetation C content averaged 44.6 ± 0.13% across all six plots. Soil P and N were also higher following fertilization in the fenced plots, where these variables were measured (Table S1). Soil C ranged from 2.49 ± 0.01% to 2.55 ± 0.01% across plots. Soil pH was 4.27 in the unfertilized plot and 4.08 in fertilized plots. Confirming reference conditions, pH measured in a single control plot in 2011 prior to fertilization was 4.21 ± 0.01.

Empirical semivariogram models show that there was a statistically significant patch structure at scales corresponding to the size of the subplots in the fenced and unfenced, heterogeneously fertilized plots (Objective 1; Figs. 3B, 3F). Also confirming expectations, in unfertilized (reference) or homogenously fertilized plots the range scale was significantly longer or shorter (Fig. 3; Table S2). The sampling distributions of the semivariogram range values for vegetation biomass determined from the maximum likelihood and bootstrapping analysis show that the range value most closely resembles that of the hypotenuse of the 2 × 2 m subplot, relative to the other subplots (Figs. 3D, 3H). Higher spatial structure in the heterogeneous versus homogeneous or unfertilized plots can also be seen in the kriged plots of biomass (Fig. 4). These maps also highlight the higher mean levels of

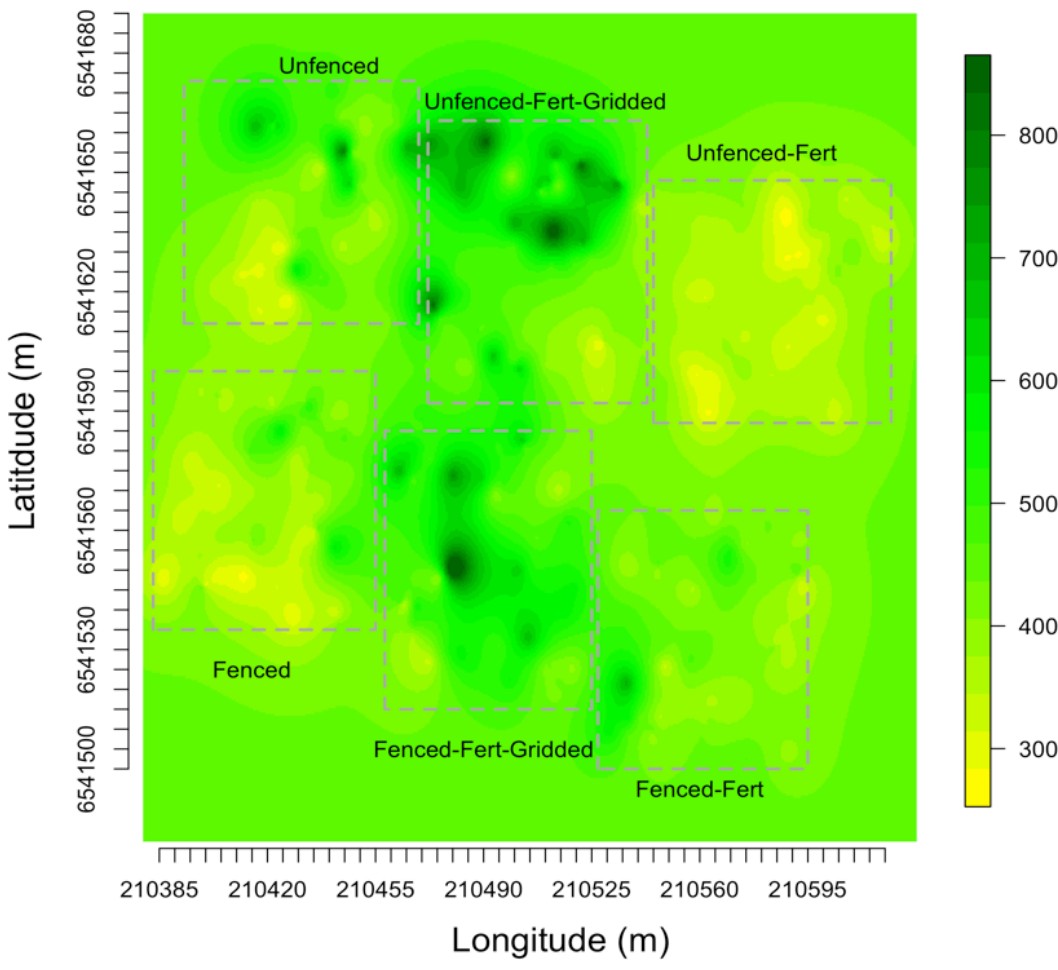

**Figure 4  Kriged biomass map.** Kriged map of biomass using ordinary kriging with a spatial covariance model optimized by a maximum likelihood analysis: (A) Unfenced, unfertilized, (B) Unfenced, heterogeneously fertilized, (C) Unfenced, homogeneously fertilized, (D) Fenced, unfertilized, (E) Fenced, heterogeneously fertilized, (F) Fenced, homogeneously fertilized.

biomass in fertilized subplots relative to areas outside of subplots or relative to other plots. These hotspots contributed to the higher than average biomass values for heterogeneously fertilized plots as a whole.

Normalized nugget/sill ratios represent the ratio of noise-to-structure in the semivariogram model, and thereby provide an estimate of the degree to which the overall variation in the model is spatially random. Nugget/sill ratios were highest in the unfenced, homogeneously fertilized plot (3.89), suggesting more random variation in the overall model variance, whereas ratios were lower (0–0.02) for heterogeneously fertilized or fenced treatments, suggesting that there was little contribution of spatially random processes in the overall model. These results support the expectation of strong spatial structure of biomass in response to nutrient addition, especially at the 2 meter scale.

The semivariogram range values for vegetation % N and % P (Table S3) were comparable to subplot scales of nutrient additions (% P, ~4.9 m, % N, ~5.8 m) in the fenced,

**Table 2  Mixed model results comparing biomass to foliar nutrients.** Results of the mixed model relating biomass to foliar nutrients, where herbivory, fertilizer type, plot treatment, and subplot size were all tested as random effects; foliar N and P represented fixed effects upon biomass, and model error was assumed to be Gaussian. A normal likelihood function was minimized to estimate optimal regression coefficients for each mixed model formulation. Both Akaike's Information Criterion (AIC) and Bayesian Information criterion (BIC) were used to compare different models. Delta ($\triangle$) represents differences in BIC between the current model and the model with the lowest BIC.

| Model | DF | AIC | BIC | Δ |
|---|---|---|---|---|
| *Random Effects* | | | | |
| **Plot** | **5** | **1092.4** | **1114.2** | **0.0** |
| Herbivore | 5 | 1190.1 | 1211.9 | 97.7 |
| Fertilizer | 5 | 1100.7 | 1122.5 | 8.3 |
| Plot \|Sub-Plot | 6 | 1090.4 | 1116.5 | 2.3 |
| Herbivore \|Sub-Plot | 6 | 1188.6 | 1214.7 | 100.5 |
| Fertilizer \|Sub-Plot | 6 | 1102.7 | 1128.8 | 14.6 |
| *Fixed Effects* | | | | |
| N + P | 5 | 1090.3 | 1112.1 | 5.3 |
| **P** | **4** | **1089.8** | **1107.3** | **0.4** |
| N | 4 | 1090.7 | 1108.2 | 1.3 |
| N : P | 6 | 1092.3 | 1118.5 | 11.6 |
| N + P + Sub-Plot | 6 | 1092.3 | 1118.5 | 11.6 |
| N + P : Sub-Plot | 8 | 1095.6 | 1130.5 | 23.6 |
| $P + N^2$ | 5 | 1091.6 | 1113.4 | 6.6 |
| $N + P^2$ | 5 | 1089.7 | 1111.5 | 4.7 |
| $N^2 + P^2$ | 5 | 1091.1 | 1113.0 | 6.1 |
| $N^2$ | 4 | 1093.3 | 1110.8 | 3.9 |
| **$P^2$** | **4** | **1089.4** | **1106.9** | **0.0** |

heterogeneously fertilized plot, where herbivores were absent. However, higher or lower range values were found for the other plots. Similar to results for biomass, the nugget:sill ratio in semivariogram models of vegetation % N and % P was highest in the unfertilized plots, suggesting a larger degree of spatially random processes contributing to overall variance. In turn, this indicates higher spatial structure captured in models of the fertilized treatments, relative to random processes. Semivariogram parameters of soil carbon and nutrients showed few differences among treatments where these were measured (fenced plots, only) (Table S3).

Mixed models used to predict biomass levels from N or P foliar concentrations, while treating plot and treatment as random effects, showed that biomass was best predicted by levels of foliar P, relative to foliar N alone or foliar N + P (Objective 2; Table 2). Although foliar P alone did better than foliar N alone as a fixed effect, the difference was marginal (<2 BIC). The 'best' model used only plot treatment type as a random effect, which outperformed model formulations using herbivory or fertilizer type and those with nested structures incorporating subplot size as random effects.

The spatial structure of heterogeneous plots was estimated to be similar between neutral and empirical semivariogram models and generally matched subplot scales

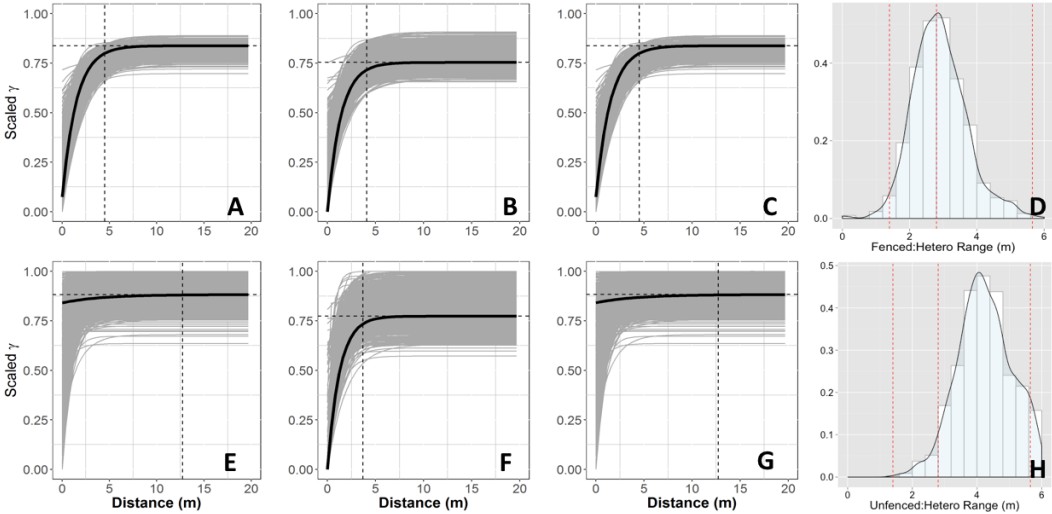

**Figure 5** **Semivariograms from neutral models.** Simulated semivariograms of vegetation biomass for each plot from neutral landscape models: (A) Unfenced, unfertilized, (B) Unfenced, heterogeneously fertilized, (C) Unfenced, homogeneously fertilized, (D) Fenced, unfertilized, (E) Fenced, heterogeneously fertilized, (F) Fenced, homogeneously fertilized. Shaded lines represent semi-variogram models fitted during the bootstrapping procedure. Dashed vertical line represents the optimal range value. Also shown: the sampling distribution of the range parameter for heterogeneously fertilized plots that were either (D) Unfenced, or (H) Fenced. The distribution was calculated with a bootstrapping approach with maximum likelihood optimization.

(Objective 3; Fig. 5). Interestingly, the neutral models estimated higher range values (longer length scales) in fenced plots compared to unfenced plots, whereas empirical semivariogram models estimated longer length scales in unfenced plots.

## DISCUSSION

Although scale-dependence is known to be critical for inferring ecological processes from ecological pattern (*Levin, 1992*; *Dungan et al., 2002*; *Sandel, 2015*), and although nutrient limitation and herbivory are known to influence grassland productivity at multiple scales (*Fuhlendorf & Smeins, 1999*; *House et al., 2003*; *Pellegrini, 2016*; *Van der Waal et al., 2016*), our study is the first to our knowledge to impose an experimental design that directly tests the scale at which grass responds to nutrient additions. By imposing the scale of nutrient additions *a priori* we were able to discern, using semivariograms based on empirical data, greater biomass response at the 2 × 2 m grain compared to finer (1 × 1 m) or broader (4 × 4 m) grain sizes. Comparisons to neutral models based on simulated landscapes with known patterns, supported our expectations that herbivore activity and nutrient additions can contribute to the spatial structure found in our empirical results. Mixed model results further indicated that foliar nutrient concentrations accounted for the majority of observed patterns in the level of biomass response, with limited influence of herbivory. Overall, these results yield data on the spatial scale of the nutrient-productivity relationship in a grassland coastal forest of the Eastern Cape, South Africa, and support the assertion that ecological processes are likely multi-scaled and hierarchical in response to nutrient additions.

### Inferring the scale of grass response to nutrient additions

This study provided an opportunity to experimentally test the scale at which nutrient limitation is most strongly expressed, providing an alternative to studies in which spatial autocorrelation is observed post-hoc. Detecting the autocorrelation structure of an ecological pattern is a critical but insufficient approach for inferring an ecological process. A preferred approach, such as tested here, is to impose a pattern at a certain (set of) scale(s) and determine if that process responds at that scale(s). The benefit to this approach is a closer union between observed responses (biomass) and ecological processes (nutrient limitation) and the ability to compare responses across scales. Our results indicate that biomass responded to nutrient additions at all subplot scales, with spatial autocorrelation of the biomass response highest at the 2 × 2 m scale. Studies have found finer-grain spatial structure in grassland soil properties (*Jackson & Caldwell, 1993a*; *Rietkerk et al., 2000*; *Augustine & Frank, 2001*) while others have observed biomass responses to nutrient additions or herbivory at finer (*Klaus et al., 2016*) or broader (*Lavado, Sierra & Hashimoto, 1995*; *Augustine & Frank, 2001*; *Pellegrini, 2016*) scales, or a limited effect of scale altogether (*Van der Waal et al., 2016*) Indeed, we observed high nugget variance for soil nutrients and carbon under heterogeneous fertilization, implying variation below the scale of sampling. The response of biomass at the 2 × 2 m scale may thus reflect spatial patterns in species composition or plant groupings rather than soil characteristics, suggesting a possible influence of competitive exclusion, at least in fenced plots where soil nutrients were sampled.

Although vegetation responses were stronger at the 2 × 2 m grain, all subplots in the heterogeneous plots responded to nutrient additions, as observed in the kriged maps. As a result, the heterogeneous plots had greater average biomass than plots which were fertilized homogeneously, despite the fact that fertilizer was added equally on a per area basis for both treatments. Several other studies have found higher biomass following heterogeneous nutrient applications. For example, *Day, Hutchings & John (2003)* observed that heterogeneous spatial patterns of nutrient supply in early stages of grassland development led to enhanced nutrient acquisition and biomass productivity. Similarly, *Du et al. (2012)* observed increased plant biomass following heterogeneous nutrient fertilization in old-field communities in China. Mechanisms for enhanced productivity following heterogeneous nutrient supply are not clear but may include shifts in root structure and function or shifts in species dominance, which were not analyzed here. For example, roots may respond to patchiness in nutrient availability by modifying root lifespan, rooting structures, or uptake rate to maximize nutrient supply (*Robinson, 1994*; *Hodge, 2004*). In turn, initial advantages afforded by plants in nutrient-rich locations may result in larger plants and advantages against competitive species, potentially via enhanced root growth (*Casper, Cahill & Jackson, 2000*).

### Implications for understanding nutrient limitations

The goal of our study was to determine the scale of grass response to nutrient additions and herbivory but our results also convey some general lessons about the role of nutrient limitation in grassland ecosystems. First, our study supports the notion of coupled N

and P limitation in grasslands (*Craine & Jackson, 2010*), including the subtropics (*Klaus et al., 2016*). *Ostertag (2010)* also showed that there was a preference for P uptake in a nutrient limited ecosystem in Hawaii and suggested that foliar P accumulation may be a strategy to cope with variability in P availability. We found that P was the variable that explained most of the variation in the level of biomass response across all plots, followed by N. In addition, we saw a strong difference in N:P ratios between reference and fertilized plots. Many studies have used stoichiometric relationships of N and P to infer nutrient limitation (*Koerselman & Meuleman, 1996*; *Reich & Oleksyn, 2004*), although there are limits to this approach (*Townsend et al., 2007*; *Ostertag, 2010*). Using this index, our N:P ratios of vegetation in reference plots would indicate co-limitation for N and P prior to fertilization (N:P > 16). Addition of dual fertilizer appeared to alleviate P limitation more than N, with N:P ratios reduced one year following treatment, indicating N limitation or co-limitation with another element (N:P < 14). Grazing may also preferentially increase grass P concentrations in semi-arid systems in South Africa (*Mbatha & Ward, 2010*) and thus the cumulative impacts of preferential plant P uptake and P additions from manure may explain the high spatial structure observed in our grazed and fertilized plots.

Relating biomass response to nutrient limitation using *in situ* data is complicated by processes such as luxury consumption (*Ostertag, 2010*), initial spatial patterns in soil fertility (*Castrignano et al., 2000*), root distribution, signaling and allocation (*Aiken & Smucker, 1996*), species and functional group shifts (*Reich et al., 2003*; *Ratnam et al., 2008*), or species' differences in uptake rates or resorption (*Townsend et al., 2007*; *Reed et al., 2012*). Spatial patterns of finer-scale processes such as microbial community composition have also been explored and are known to influence rates of nutrient cycling (*Ritz et al., 2004*; *Smithwick et al., 2005*). In the case of heterogeneous nutrient supply, species competitive relationships across space may be enhanced (*Du et al., 2012*) and may result in increases in plant diversity (*Fitter, 1982*; *Wijesinghe, John & Hutchings, 2005*), although other studies have found little evidence to support this claim (*Gundale et al., 2011*). Together, these factors may explain any unexplained variance of vegetation N and P concentrations that we observed. Shifts in species composition were likely minimal in this study given the short-term nature of the study (one year), but patchiness in biomass responses indicate size differences that could have modified competitive relationships in the future (*Grime, 1973*). Unfortunately, the site burned one year following the experiment, precluding additional tests of these relationships.

## Herbivory-nutrient interactions

Our study indicates a strong scalar influence of nutrient additions relative to nutrient-herbivore interactions. First, we found that the significant length scale was similar between unfenced and fenced plots, indicating that herbivory did not alter the grain of biomass response to nutrient limitation. In addition, herbivory was not significant in final mixed effects models, relative to the inclusion of foliar nutrient variables, suggesting that nutrients had a greater influence on the level of biomass response. However, our study was not designed to unravel the multivariate influence of herbivores on grasslands, which may influence vegetation biomass through biomass removal, movement activity, and manure

additions (*Milchunas & Lauenroth, 1993*; *Adler, Raff & Lauenroth, 2001*; *Van der Waal et al., 2016*). Interestingly, our empirical semivariogram model indicates longer range scales where herbivores were present compared to simulated semivariogram models, which may reflect homogenization of biomass through grazing and thus a greater top-down approach of herbivory on ecosystem productivity than previously appreciated (*Van der Waal et al., 2016*), or other complex interactions between grazing and fertilization not accounted for in the current study.

### Uncertainties

There are several key uncertainties and caveats in applying our methodological approach more broadly. First, the experimental design described herein was labor-intensive, requiring both precision mapping of locations for nutrient additions and post-treatment vegetation sampling, as well as extensive replication of treatments that would respond to broader ecological patterns, i.e., grazing. This necessitated a trade-off between sampling effort across scales (subplots, plots). Important processes at scales above and below the extent and grain of sampling used here were likely important but were not included. Second, our neutral models assumed additive effects of herbivore activity and fertilization; in contrast, empirical results likely reflect complex, potentially non-additive, interactions between grazing and fertilization. Third, recent work has suggested that both nutrient patchiness and the form of nutrient limitation (e.g., N vs. P) may change seasonally (*Klaus et al., 2016*), which was not assessed here. Moreover, annual variation in precipitation, in our case a dry year followed by a wet year, may have influenced the level of biomass response to nutrient additions.

## CONCLUSIONS

Understanding the factors that regulate ecosystem productivity, and the scales at which they operate, is critical for guiding ecosystem management activities aimed at maintaining landscape sustainability. New approaches are needed to characterize how ecosystems are spatially structured and to determine whether there are specific scale or scales of response that are most relevant. In South Africa, grasslands cover nearly one-third of the country and maintain the second-highest levels of biodiversity but are expected to undergo significant losses in biodiversity in coming decades due to increasing pressure from agricultural development and direct changes in climate (*Biggs et al., 2008*; *Huntley & Barnard, 2012*). We employed a neutral model approach to test for ecological process, an approach that has been advocated for decades (*Turner, 1989*) but which is rarely imposed (but see *With, 1997*; *Fajardo & McIntire, 2007*). We conclude that these grasslands express nutrient limitation at intermediate scales ($2 \times 2$ m) and exhibit relatively strong nutrient limitations for both N and P, with a more limited influence of herbivory. By extending this approach to other areas and other processes, specifically by imposing experimental studies to test for the influence of scale on other ecological processes, it may be possible to reduce bias in empirical studies, minimize the potential for scale mismatches, and deepen insights into ecological pattern-process interactions.

## ACKNOWLEDGEMENTS

We are deeply grateful to the support of Jan Venter, the Eastern Cape Parks and Tourism Board, and especially the Mkambathi Nature Reserve for allowing the establishment of the "Little Pennsylvania" study area. We are grateful to the hard work of the Parks and People 2010, 2011, and 2012 research teams, which included undergraduate students from The Pennsylvania State University who helped in all aspects of field research. Special thanks to the dedication of Sarah Hanson, Warren Reed, Shane Bulick, and Evan Griffin who assisted tirelessly with both field and laboratory work. We also thank the reviewers and editors for their insightful comments.

### Funding

This research was made possible through support from the National Science Foundation-Division of Environmental Biology grant (NSF-DEB 1045935) and a National Science Foundation-Research Experience for Undergraduates grant (NSF-DEB 1045935) to EAH Smithwick, and the Penn State College of Earth and Mineral Sciences George H. Deike, Jr. Research Grant. The funders had no role in study design, data collection and analysis, decision to publish, or preparation of the manuscript.

### Grant Disclosures

The following grant information was disclosed by the authors:
National Science Foundation-Division of Environmental Biology: NSF-DEB 1045935.
National Science Foundation-Research Experience for Undergraduates: NSF-DEB 1045935.
Penn State College of Earth and Mineral Sciences George H. Deike, Jr. Research Grant.

### Competing Interests

The authors declare there are no competing interests.

### Author Contributions

- Erica A.H. Smithwick conceived and designed the experiments, performed the experiments, analyzed the data, wrote the paper, prepared figures and/or tables.
- Douglas C. Baldwin performed the experiments, analyzed the data, prepared figures and/or tables, reviewed drafts of the paper.
- Kusum J. Naithani analyzed the data, prepared figures and/or tables, reviewed drafts of the paper.

### Field Study Permissions

The following information was supplied relating to field study approvals (i.e., approving body and any reference numbers):

South Africa Eastern Cape Parks and Tourism Agency, Permit #RA0081.
USDA APHIS Permit to import plant species for research purposes PDEP11-00029.

## Data Availability

The raw data has been supplied as a Supplemental Dataset.

## Supplemental Information

Supplemental information for this article can be found online at http://dx.doi.org/10.7717/peerj.2745#supplemental-information.

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
