# Peer review of "Grassland productivity in response to nutrient additions and herbivory is scale-dependent"

_PeerJ, doi:10.7717/peerj.2745_

## Round 0.1 · original submission · Major Revisions

· Academic Editor

Major Revisions

Useful comments from the reviewers. Please consider addressing these ideas.

·

Basic reporting

no comments

Experimental design

no comments

Validity of the findings

no comments

Comments for the author

The manuscript by Smithwick is an attempt to infer correctly the effects of fertilization and grazing on grass biomass in an experiment. And I say “correctly” because they consider the spatial autocorrelation associated with the factors involved in the response of plants, especially when experiments are of medium to large–scale. For this they computed the variance associated with spatial autocorrelation via empirical and neutral semivariogram models. In general, I think the purpose of the study is well justified and I am for the publication of this manuscript. However, the authors need to do major work in the way the manuscript is presented. The manuscript suffers clarity because it is not an ecological process–driven only study, and it is not a methodological only study but a mix, and therefore it is not the easiest to write about. And this confusion is patent. I would recommend the authors to just follow the title they have given it to the study. To say, start telling the story in ecological terms, the whys nutrient can affect grass biomass and what is known and what is not known; then to introduce the “spatial scale” problem and how it can actually be taking into account to improve inference. In this way, you introduce your study from an ecological point of view (more understandable, easier to follow and also with greater impact) and then add the methodological aspect of all this (the how we win following your approach). Although I have said the title is fine, I think it can be improved by showing some more affirmative approach, like: nutrient limitation in grasses is more (or less) obvious when considering the spatial pattern structure: an empirical and neutral semivariogram approach.

Other minor issues:
L19, "this": ambiguous pronoun reference; add noun or phrase or rephrase for clarity.
L37, so, this is methodological study with no ecological relevance, at least in the summary...
L68, see also McIntire (2004), McIntire & Fajardo (2009), and Fortin & Gurevitch (2001)
L76, “…challenging THE understanding…”
L89,”…chosen TO correspond…”
L127, in comparison with natural systems, productive systems will always be nutrient limited...
L224, you need to be more specific here. My recount is that you are fitting empirical semivariograms to, among other things, estimate the range of the semivariogram, which gives you an idea of the spatial (scale) structure of the data. Then you talk about the sill and nugget but never refer to the empirical semivariogram...
L229, I would recommend to define first the semivariogram parameters (nugget, sill and range)
L267, this may be similar to what Fajardo & McIntire (2007) did:
L279, how do the semivariograms look like? What are the parameter values?
L284, I don't get it; could you please be more specific?
L327, this expectation would be better placed earlier in the text, in the section of methods regarding objective 1
L331, I insist, all this necessary text belongs to the methods, there I missed a more appropriate treatment of the expectations, where they should be tightened to semivariogram parameters (values).
L332, here it is where indeed the results start…
L343, you need to explain what a nugget:sill ratio stands for! A nugget:sill ratio indicates how much of overall variation is spatially random (according to what the nugget means). In the text, there are many parts that have been assumed to be known by the reader but these assumptions are hastened...
L350, or the nugget:sill ratio
L351, not sure about what you are saying here: I think that the higher the nugget variance (and nugget:sill ratio) the lower the spatial structure of a variable...
L365, although true, these interpretations of results are more appropriate for the Discussion section
L390, I think it is better to start the Discussion section introducing THE result of your study and then commenting it and describing why your results are important... When is not like it, it reads as Introduction...
In general, de Discussion section does not follow a clear order. Please, discuss your results matching your three objectives. Perhaps, this lack of order comes from the Introduction section (see above).
L491, conclusions are not a summary of results nor a repetition of the methodological merits but a section where you can write more general things about your study, what is beyond it (for other disciplines, maybe) and what other ecological processes can gain from your results. Also, Conclusions are intended to identify future work in the subject matter.

Revised by Dr. Alex Fajardo.

References
Fajardo, A. and E. J. B. McIntire. 2007. Distinguishing microsite and competition processes in tree growth dynamics: an a priori spatial modeling approach. The American Naturalist 169:647-661.
Fortin, M.-J. and J. Gurevitch. 2001. Mantel Test. Spatial structure in field experiments. Pages 308-326 in S. M. Scheiner and J. Gurevitch, editors. Design and analysis of ecological experiments. Oxford University Press, New York.
McIntire, E. J. B. 2004. Understanding natural disturbance boundary formation using spatial data and path analysis. Ecology 85:1933-1943.
McIntire, E. J. B. and A. Fajardo. 2009. Beyond description: the active and effective way to infer processes from spatial patterns. Ecology 90:46-56.

Reviewer 2 ·

Basic reporting

This article presents the results of an experiment designed to estimate the effects of scale on biomass and nutrient concentration of grasslands in relation to nutrient addition and herbivory. The Authors utilize neutral models to contrast against the observed spatial patterns and determine the scale of response. This is a very comprehensive study that is overall well designed and analyzed. My main concern about basic reporting for this article is in the Introduction. The conceptualization of scale is relatively light and aspects such as factors determining spatial patterns or the approaches currently used to determine the effects of scale on ecosystem properties should be better reviewed. Also, the effects of herbivory and how these relate to scale are not well set up as a theoretical background. I also find that the literature in the MS is a bit dated. The third objective as proposed is not possible to be tested with this design using sites in a single environment. I suggest merging Objective #3 with Objective #1 as both objectives use the same response variables, and leave Objective #3 as a point in the Discussion.

In the Results section, some paragraphs repeat information presented in either the Introduction or Methods and hence those statements should be removed, especially at the beginning of paragraphs. L307-318 repeat, at times, information presented in Table 1. Additionally, L365-373 should be moved to the Discussion. The inclusion of herbivory as a factor of discussion is not clear, although it is included in the neutral models. L365-373 do address herbivory, but require expansion when they are moved to the Discussion. L468-485 that discuss aspects related to grassland conservation should be reduced as they do not relate directly to the main focus of the article. Instead, I suggest adding a paragraph with advantages/disadvantages of the method proposed in the MS, i.e. combining a field study with neutral models to test scale effects.

As a final note on the Abstract/Title, there is no indication of the use of fencing (i.e. herbivory), which is the other main factor analyzed in this MS. Please modify accordingly.

Experimental design

As I stated previously I believe this is an overall well designed and analyzed study. My only concern in this section is on the selection of the six main plots (was it random?) and whether there is information on baseline nutrient content for the plots that received nutrient addition. Are all six plots equal in terms of soil nutrient composition? This information can be added as an Appendix. This is important as the responses reported in this MS were collected after a year, so the baseline conditions will have an important effect on the responses of the vegetation. Also, it would be advisable to get an estimation of the weather characteristics of the year, was it a dry year? a rainy year? This information should be added to the Methods section.

Validity of the findings

My only comment in this section is in relation to the third objective. As stated before, I believe that objective should be merged with Objective #1, and the justification/interpretation only considered in the Discussion.

Comments for the author

In this section I only have a few minor editorial comments:
L59-61, how is bark beetles related to grasslands? Please remove.
L72, replace “In so doing” with “in doing so”
L106, comparable or similar?
L150, please add a citation for those figures.
L178, GPS brand, and model, precision?
L182, both compounds were equally applied? Or which plot received what? When were the nutrients added? Before the growing season?
L221, supplementary, not supplmentary.

Also, I hope the Authors find these references useful:
Dungan, et al. 2002. A balanced view of scale in spatial statistical analysis. Ecography 25: 626-640.
Sandel, B. 2014. Towards a taxonomy of spatial scale dependence. Ecography 38: 358-369.

---

## Round 0.2 · Minor Revisions

· Academic Editor

Minor Revisions

Please address the comments provided including these two: I recommend the Authors to properly conceptualize these terms in order to correctly set-up their study. I also recommend the Authors to include more ecological explanations of the spatial processes they refer to.

Reviewer 2 ·

Basic reporting

This a revision of a previously submitted MS that evaluates the effects of nutrient addition and herbivory on biomass and nutrient concentration of a coastal grassland system across different scales, focusing on the later concept (scale-dependency) via an experimental approach. I find this is a very much improved version of the MS and one that is almost ready for publication. I thank the Authors for addressing my comments and congratulate them for their work. I do, however, have some minor comments regarding basic reporting of their MS. Two of the most used concepts in their MS are grain and extent (initially used in L46), yet these two remained properly undefined throughout the MS. I recommend the Authors to properly conceptualize these terms in order to correctly set-up their study. I also recommend the Authors to include more ecological explanations of the spatial processes they refer to. This is not a major issue, but for example some sections in the Introduction could use this, for example L126-129, L145-147, or L150-153. Finally, the Abstract neglects setting-up the stage for the herbivory treatment, which is something that should be done in order to show the full scope of their MS.

Experimental design

I have no more comments in this section as I am happy with the revisions made by the Authors.

Validity of the findings

I have no more comments in this section as I agree with Authors´ response.

Comments for the author

An overall comment on the MS is to give it a good read as it still has some minor typological/grammar mistakes, for example in L45 where is an extra "and", L69 where the word "patterns" seems out of context; or some awkward wording such as L443-445, where the sentence starts using a negative on the purpose of the study. This is not an exhaustive list, but I suggest the Authors to take a closer look at their MS to avoid these.

---

## Round 0.3 · accepted · Accept

· Academic Editor

Accept

Thank you for taking the time to address the comments.